# Analysis of Acoustic Emission Characteristics and Failure Mode of Deep Surrounding Rock of Sanshandao Gold Mine

**DOI:** 10.3390/ijerph192013351

**Published:** 2022-10-16

**Authors:** Guang Li, Rong Lu, Fengshan Ma, Jie Guo

**Affiliations:** 1Key Laboratory of Shale Gas and Geoengineering, Institute of Geology and Geophysics, Chinese Academy of Sciences, Beijing 100029, China; 2Innovation Academy for Earth Science, Chinese Academy of Sciences, Beijing 100029, China; 3Beijing Urban Construction Design & Development Group Co., Ltd., Beijing 100029, China

**Keywords:** fractured rock, acoustic emission, parameter analysis, *b*-value, failure mode

## Abstract

In mining engineering, crack distribution has a considerable influence on the mechanical behavior and stability of the surrounding rock mass. Using the granite of the Sanshandao gold mine as experimental samples, the deformation and failure of fractured rock were analyzed based on a rock uniaxial compression test with acoustic emission monitoring. We analyzed the characteristics of different stages of rock sample deformation, and evaluated the failure mode of seven types of rock samples. The results show that the cracks had a considerable impact on rock sample strength and mechanical behavior, and the strength of intact rock was the highest, while that of the sample with parallel double cracks was the lowest. The acoustic emission parameters, AF, RA, and lg(AF/RA), have different change trends in different stages of rock deformation and failure. Based on these change trends, the failure modes of rock samples with different crack distributions were identified. Additionally, for the rock samples with seven types of crack distribution, a sudden or progressive failure caused by the *b*-value curves was observed. The research findings provide a database for deep surrounding rock stability in the study area and provide suggestions for failure prediction.

## 1. Introduction

Due to economic growth, the need for resources, especially mineral resources, is increasing. The depth of mining engineering has also been increasing [1,2]. As the depth of mining increases, rock deformation and the failure of deep roadway become the main causes of mining safety accidents [3,4]. In most metal mines, hard rock with cracks is a major component of surrounding rock mass. The existence of cracks can seriously affect the strength and deformation failure characteristics of rock and is often the main cause of rock instability. Therefore, it is of great significance for the stability of mining engineering to understand the damage characteristics and failure mode of deep fractured rocks [5,6,7].

Some studies on surrounding rock failure use methods such as experimental tests, mechanical analyses, acoustic emissions, and numerical simulation [8,9,10,11,12]. Among them, acoustic emission (AE) could provide information about the development of cracks, and is helpful for understanding and predicting rock mass failure [13,14,15,16,17]. Relevant studies using AE have been conducted in recent years. Holcomb et al. studied the acoustic emission characteristics of granite under uniaxial compression, and the results show that the damage surface of isotropic materials under uniaxial compression was highly anisotropic [18]. Rao et al. used ultrasonic and acoustic emissions to study the failure process of brittle rocks under cyclic loading [19]. Lockner discussed the role of acoustic emission in the study of rock fracture failure mechanism [20]. Mansurov et al. studied the acoustic emission characteristics of microcrack accumulation and breakthrough cracks in the process of rock failure, and predicted the type of failure according to the acoustic emission characteristics [21]. Rudajev et al. studied the autocorrelation characteristics of the acoustic emission parameter time series of granite through experiments, and proposed the precursor characteristic parameters that can effectively predict rock failure [22]. Chang et al. defined the damage thresholds before the peak strength of Hwangdeung granite and Yeosan marble under triaxial compression by the moving point regression technique using acoustic emission data [23]. He et al. studied the characteristics of AE during the rock burst process of limestone under true triaxial unloading conditions [24]. Patricia et al. studied the acoustic emission characteristics of granite and marble cylinder specimens under diameter-specific loading conditions, and the results show that the microcracks first appeared at 90% and 60% of the peak strength, respectively [25]. Xiao et al. studied the relationship of acoustic emission characteristics and the stress release rate of coal samples in different dynamic destruction stages [26]. Agioutantis et al. studied the acoustic emission characteristics of marble under scatter loading and discussed the possibility of acoustic emission in rock failure prediction [27]. Yang et al. explored the acoustic emission behavior of granite after subjection to different high-temperature treatments [28]. 

In conclusion, AE characteristic parameters can reflect the dynamic process of crack development under loading. However, previous research mainly focused on the experimental study of intact specimens or samples with single joints, and few studies considered the interaction between combinations of different fractures. In order to fully understand the failure characteristics of fractured rock mass, it is necessary to first clarify the connection mode between the joints. Hence, taking the granite of the Sanshandao gold mine as experimental samples, the AE characteristics, crack propagation, and rock failure of seven specimens with different fracture distributions were analyzed. The research findings provide a database for deep surrounding rock stability in the study area and suggestions for failure prediction.

## 2. Geological Environment and Engineering Properties

### 2.1. Geological Environment and Field Investigation of Study Area

Sanshandao Gold Mine, located on the Jiaodong Peninsula, is the only coastal bedrock metal mine that still operates in China (Figure 1a). The alteration and mineralization of the Sanshandao gold deposit are widely developed. The hydrothermal alteration is mainly potassic, silicified, sericite, pyrite, and carbonated. The mineralization styles are mainly altered rock disseminated type and quartz vein type. The ore body is 900 m in length and 25 m thick, located in the fault zone of F1 (Figure 1b). The average strike and dip angles of the ore body are 40° NE and 35–50°, respectively. The fault zone strikes 35–40° NE and gradually steepens from NE to SW, with the average dip angle increasing from 46° to 70° (Figure 1c). The dip direction is SE. The thickness of the ore body varies between 0.95 and 12.18 m, and the average thickness is 10.42 m. The exposed strata in the area are mainly Quaternary sediments and granite of the Jiaodong Group. The Quaternary layers are the most widely distributed strata in the area. Thicknesses range from 8 to 10 m, but the maximum is over 60 m, consisting mainly of medium-coarse sand, sandy clay, clayed sand, medium-coarse sand with gravel, and clay layers from top to bottom. The bedrock of the deposit is mainly composed of metamorphic and magmatic rocks. The alteration zone has been obviously zoned through multi-stage tectonic movement, and it basically spreads along the tendency and strike of fault F1. The gold deposits mainly occur in the cataclastic rocks within 40 m of fault F1, and they mainly consist of veined and reticulated mineralized rocks [29,30,31,32,33].

The mechanical upward horizontal slice stopping–filling method with pointed pillars is applied in the mining areas. The stope length is 100 m, and its width is the thickness of the ore body. The heights of the sublayer, cutting layer, and filling body are 3, 4.5, and 3 m, respectively. Horizontal cut and fill mining is adopted for some stopes [34]. At present, the mining level of the study area is nearly 1000 m. The in situ stress field is composed of gravitational and tectonic stress fields. By measuring the in situ stress, the law of maximum horizontal principal stress, the minimum horizontal principal stress, and the vertical principal stress can be determined, as shown in Equations (1)–(3) [35].
(1)$$\sigma_{\mathrm{hmax}}=0.11+0.0539z$$
(2)$$\sigma_{\mathrm{hmin}}=0.13+0.0181z$$
(3)$$\sigma_{z}=0.08+0.0315z$$

In the equations, *z* is mining depth (m), and the unit of *σ*_hmax_, *σ*_hmin_, and *σ_z_* is MPa.

### 2.2. Drilling Investigation and Survey of Field Cracks of Surrounding Rock Mass

#### 2.2.1. Drilling Investigation of Study Area

The depth of the drilling cores is −2000 m, and the drilling operation determined the characteristics of deep rock cracks. In the process of drilling, the vertical stress of rock core was initially relieved, and the large stress difference disappeared until the horizontal stress was relieved. The stress difference resulted in rock damage. During the investigation, a high ground stress was proven by the formation of disk-shaped rock cores in every drilling hole in the study area, as shown in Figure 2. The major cracks of deep drilling cores at −2000 m were recorded via statistical analysis. The dip angle of the major crack is between 30° and 60° [36].

#### 2.2.2. Field Investigation of Surrounding Rock Cracks in a Deep Roadway

The surrounding rock of the metal mine is hard rock with cracks. In the process of mining, surrounding rock deformation is inevitable. Surrounding rock cracks in the study area were investigated in mining engineering, as shown in Figure 3. Anisotropy and heterogeneous properties of rock mass can give rise to stress concentrations in the loading process. Crack initiation caused by stress concentration is the major cause of rock mass deformation and failure. Rock mass crack distribution, location, and combination type influence their initiation and propagation.

## 3. Experimental Results and Analysis

### 3.1. Rock Samples Description

Rock samples in the following tests were all granite taken from drilling exploration cores in the Sanshandao gold mine. Granite with suitable integrity and homogeneity was used, with a density of 2580 kg/m^3^. A series of core slices were identified using optical microscopy. The rock sample composition is approximately 34% plagioclase, 27% potassium feldspar, 30% granular quartz, and 6% flaky biotite, with accessory minerals such as calcite, white mica, brown epidote, etc., as shown in Figure 4 [36].

Rock cores were processed into seven groups of cuboid samples, and the sample heights were 100 mm × 50 mm × 30 mm (length × height × thickness). According to the investigation of cracks in the roadway, the dip angle of rock sample cracks was set as 60°. A cutting machine was used to precast cracks in the granite specimens. The length of the fractures was 15 mm, and the width was 1 mm. The crack distributions of rock samples were divided into seven types, including no crack, single crack, parallel double cracks, collinear double cracks, conjugate double cracks, intermittent double cracks, and cross double cracks. The rock samples are shown in Figure 5. In order to avoid the influence of accidental errors, three identical samples were made for each group, and repeated experiments were carried out.

### 3.2. Test Method

A uniaxial compressive test was conducted on the WES-2000 digital hydraulic servo testing machine, and the uniaxial compressive strength (UCS) was recorded. The loading speed was set as 0.02 kN/s, and the loading was applied until the sample was destroyed. A PCI-2 acoustic emission system was used to collect the acoustic emission data. In order to not affect the crack development observation, acoustic emission sensors were attached to the two sides of the model. The preamplifier gain was set to 40 dB, the threshold was set to 50 dB, and the sampling rate was set to 2.5 MHz. The boundary conditions of the model samples are shown in Figure 6.

The damage evolution characteristics of granite belonging to the Sanshandao gold mine were analyzed by AE signals, and the intact rock sample is evaluated. In the process of the uniaxial compression test for intact rock mass, three parameters—AF, RA, and lg(AF/RA)—were adopted to analyze the rock mechanical behavior. These three parameters indicated the failure characteristics of the rock mass with different types of cracks. The calculation methods of AF and RA are shown in Equations (4) and (5) [37,38]:(4)AF=Ring count/Duration time
(5)RA=Rise time/Amplitude

A large number of experiments from previous studies showed that the AF value is large when tensile cracks occur in the rock sample [39,40]. Additionally, the RA value is large when shear cracks occur in the rock sample. The parameter lg(AF/RA) represents the overall failure characteristics of the rock sample. Some researchers suggest that the process of crack initiation and propagation of the granite rock sample is divided into eight stages: crack closure, line elastic deformation, micro-crack initiation, stable micro-crack growth, micro-crack coalescence, unstable macro-crack growth, macro-crack coalescence, and failure [41,42].

### 3.3. Test Results

#### 3.3.1. Uniaxial Compressive Strength of the Samples

The rock strength testing is shown in Figure 7. The strength of the intact rock sample is 148.7 MPa. Additionally, the strength of the rock sample with a single crack is 109.8 MPa, and the strengths of rock samples with parallel double cracks, collinear double cracks, cross double cracks, conjugate double cracks, and intermittent double cracks are 96.4 MPa, 103.7 MPa, 110.2 MPa, 104.1 MPa, and 104.2 MPa, respectively. The strength of the intact rock sample is the largest value in the tests. Additionally, the strength of the rock samples with parallel double cracks is the lowest. The number and distribution of rock sample cracks have a substantial impact on rock sample strength and mechanical behavior.

#### 3.3.2. AE Characteristics Analysis of Intact Rock Sample

The failure process of intact rock was used as an example for analysis, and Figure 8 shows the curves of AF, RA, and lg(AF/RA). Combined with the running count for acoustic emission monitoring, the intact rock sample failure process was studied. At the stage of crack closure, the value of RA is small and gradually changes. In the meantime, the value of AF slightly increases. The curves demonstrate that, in the initial stage, the shear behavior of internal particles of rock sample slightly increases. With the increase in external loading, the rock sample deformation reached the stage of line elastic deformation. The value of AF sharply increased. The curves indicated that the rock sample was squeezed. At the same stage, the value of RA slowly changed, and the secondary cracks did not appear. Then, the cracks in the rock samples started their initiation and steadily developed. The rock sample developed to the stage of stable micro-crack growth. The value of AF significantly decreased, and the value of RA increased with a significant fluctuation. The value of lg(AF/RA) was stable, and the running count slightly increased at this stage. This indicated that the shear behavior of the rock sample was apparent, and shear micro-cracks were created. As the external load continued to increase, the rock sample deformation reached the stage of unstable macro-crack growth. The value of RA continued to fluctuate in a large value area. The shear cracks were created and propagated. The value of AF and running count sharply increased at this stage, demonstrating that the tensile cracks also propagated. The value of lg(AF/RA) reached a peak and then decreased, indicating that the macro-cracks were formed. In this stage, the strength of the rock sample decreased. Tensile shear composite failure occurred in the intact rock sample.

#### 3.3.3. Comparative Analysis of Acoustic Emission Signals

A comparative analysis was conducted on rock samples with a single crack and intact rock samples. As shown in Figure 9, in the initial stage of deformation of rock samples with a single crack, the value of RA remained stable. The curves demonstrated that the internal particle of rock samples with a single crack presented little shear behavior in the stages of crack closure and line elastic deformation. The value of the AF parameter fluctuated at a certain value in a small range throughout the entire process. Additionally, the value of lg(AF/RA) of the rock sample with a single crack is evidently larger than the intact rock sample. Finally, tensile failure occurred in the rock sample with a single crack and slight shear behavior.

The curve parameters of rock samples with double cracks are shown in Figure 10. The mechanical behavior of rock samples with parallel double cracks is similar to rock samples with single cracks. Tensile behavior is more evident than shear failure. The major macro-cracks of rock samples with parallel double cracks was produced by tensile behavior. The failure mode of rock samples with conjugate double cracks is tensile–shear composite failure. The value of RA was large in the process of rock deformation, which indicated that the shear behavior occurred at the initial stage. 

In conclusion, tensile cracks and shear cracks resulted in rock sample failure. The results of the failure mode of other rock samples with double cracks were concluded. Tensile–shear composite failure occurred in the rock samples with cross double cracks. The shear behavior was evident in the whole process of deformation. The characteristics of curves of the rock sample with collinear cracks are similar to the rock sample with intermittent double cracks. Compared with curves of the intact rock sample, the shear behavior of these two rock samples was slight, and the tensile behavior was strengthened. Finally, tensile failure occurred in the rock samples with collinear and intermittent double cracks.

## 4. Discussion

The *b*-value, as an important parameter of AE signal, represented the features of micro-crack development. Lei et al. [43] carried out several triaxial tests and indicated that the *b*-value could serve as a precursor to the instability failure of rock specimens. Amitrano [44] demonstrated the characteristics of the *b*-value using numerical simulations. The change in the *b*-value is closely related to the brittleness of the rock. The *b*-value change is sensitive to the unstable growth of macro-cracks [45,46,47,48,49]. Therefore, the *b*-value is employed as an early warning signal of rock failure. However, the current related studies concentrated on materials that have a strength of less than 100 MPa, such as coal and rock-like material. In this article, a *b*-value analysis could provide an indication of high-strength rock mass [50,51,52,53,54].

The *b*-value curves of seven groups of rock samples are shown in Figure 11, Figure 12 and Figure 13. The *b*-values of the intact rock sample and a single-crack rock sample range from 0.15 to 0.6. Additionally, the *b*-values of the other rock samples with double cracks range from 0.025 to 0.25.

According to the stage division, as previously mentioned, the *b*-value has different change trends in different stages. For the intact rock sample, the *b*-value fluctuated between approximately 0.05 and 0.1 in the crack closure stage, as shown Figure 11a. Then, the *b*-value sharply increased. Subsequently, the *b*-value showed an upward trend in the elastic deformation stage and reached a peak in micro-crack initiation stage. In the crack initiation stage, the rock sample was dominated by small-scale cracks. The intact rock sample deformation entered the unstable macro-crack growth stage, and the local large-scale cracks inside rocks caused severe damage, corresponding to a rapid decline in the *b*-value. Local cracks propagated following previous cracks and the *b*-value continued to decrease. The failure of the intact rock sample suddenly occurred, and rock sample damage occurred following sound that caused a part of the rock to fall. The rock sample was not intact, as shown in Figure 11b.

For the rock sample with a single crack, the *b*-value fluctuated between approximately 0.04 and 0.06 in the crack closure stage, as shown Figure 12a. Then, the *b*-value sharply increased to a peak at the stage of micro-crack initiation. Subsequently, the *b*-value gradually decreased. Finally, the rock sample with a single crack showed progressive failure, and the rock sample remained intact, as shown in Figure 12b.

A comparison analysis of *b*-values was conducted between intact rock and rock samples with a single crack. The curves of the *b*-value could be divided into two types of failure curves: a sudden failure curve (*b*-value curve of intact rock) and a progressive failure curve (*b*-value curve of rock sample with a single crack). The characteristic of the sudden failure curve is that the *b*-value reached a peak in the unstable macro-crack growth stage. The *b*-value could predict rock failure to some degree. Additionally, the characteristic of progressive failure curve is that the *b*-value reached the peak in the end-of-line elastic stage. Additionally, in the following stages, the *b*-value gradually decreased. This type of *b*-value could not predict rock failure.

Figure 13a–j shows the *b*-value curves and failure modes of rock samples with double cracks. The *b*-value peak of rock samples with double cracks was smaller than the intact rock and rock samples with single cracks. According to changing trends of the *b*-value curves, the rock samples with double cracks could also be divided into two groups: sudden failure and progressive failure. The rock samples with cross double cracks, collinear double cracks, and conjugate double cracks matched the characteristics of sudden failure. The integrity of these three rock samples was compromised. The rock sample with parallel double cracks and intermittent double cracks matched the characteristic of progressive failure. These two rock samples remained intact.

## 5. Conclusions

(1) Comparing the mechanical test results of the seven specimens with different fracture distributions, the strength of the intact rock is the highest, while that of the samples with parallel double cracks is the lowest. The number and distribution of rock sample cracks have significant impacts on rock sample strength and mechanical behavior.

(2) The acoustic emission parameter analysis showed that tensile–shear composite failure occurred in the intact rock sample. Tensile failure with slight shear behavior occurred in the rock samples with single cracks. The mechanical behavior of rock samples with parallel double cracks is similar to the rock samples with single cracks, and tensile behavior is more evident than shear behavior. The tensile–shear composite failure occurred in the rock samples with conjugate and cross double cracks, and tensile failure occurred in the rock samples with collinear and intermittent double cracks.

(3) According to changing trend of *b*-value curves, the intact rock sample suddenly failed, and the rock samples were damaged by sound and parts of the rock falling. The rock samples with single cracks showed progressive failure, and the rock samples remained intact. The rock sample with cross double cracks, collinear double cracks, and conjugate double cracks matched the characteristics of sudden failure. The integrity of these three rock samples was compromised. The rock sample with parallel double cracks and intermittent double cracks matched the characteristic of progressive failure.

(4) The mechanical behavior of rock mass not only depends on the mechanical properties of the rock itself, but also on the properties of discontinuous structural planes contained in the rock mass. The joints not only reduce the strength of the rock mass and strengthen the anisotropy of mechanical parameters, but also play a leading role in the failure mode of rock mass. Fractured rock mass is a complex engineering medium often faced in underground engineering. Our experimental study on mechanical properties and failure modes of rock mass specimens containing multiple joints not only provides technical support for the design and construction of underground caverns, but also provides an empirical reference for numerical simulations.

## Figures and Tables

**Figure 1 ijerph-19-13351-f001:**
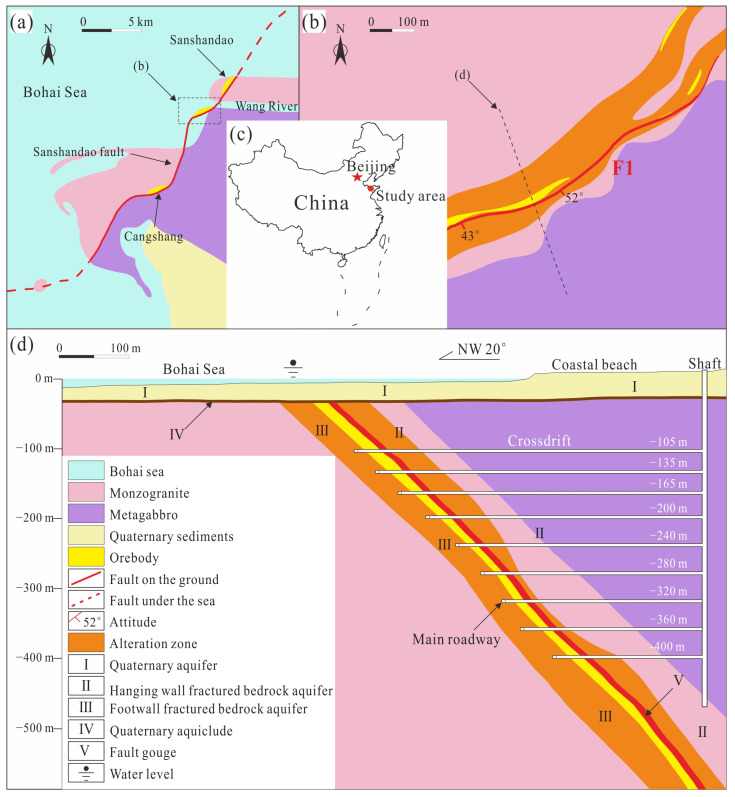
Engineering geological map of Sanshandao Mine. (**a**) Simplified geologic map showing the Sanshandao fault and other features; (**b**) close-up geologic map of the Xinli gold deposit area showing the fault and cross-section location; (**c**) location of the study area in China; (**d**) cross-section showing the fault, geology, and hydrogeologic features, as well as the underground workings of the Xinli gold deposit.

**Figure 2 ijerph-19-13351-f002:**
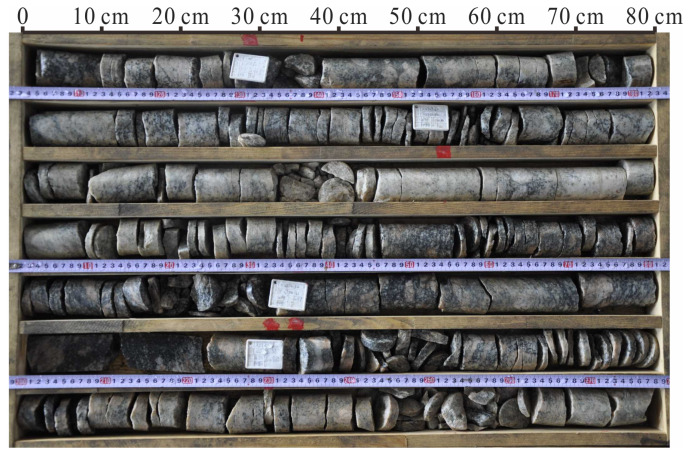
Drilling cores of the study area.

**Figure 3 ijerph-19-13351-f003:**
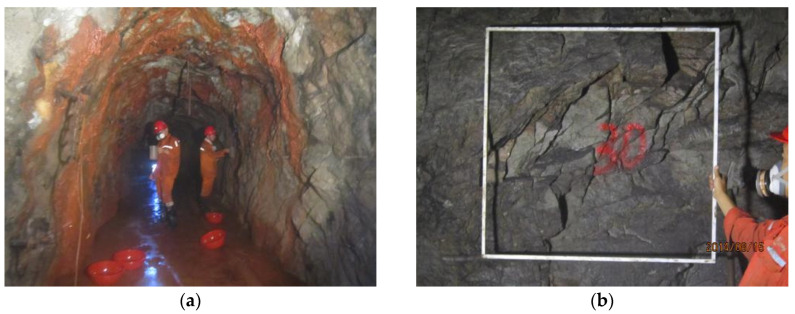
Investigation of roadway in deep mining. (**a**) Field survey; (**b**) fracture statistics.

**Figure 4 ijerph-19-13351-f004:**
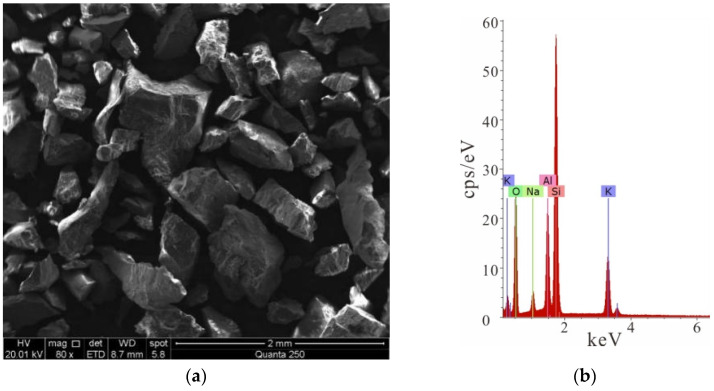
Microscopic scanning of rock sample. (**a**) Microscopic scanning of rock sample; (**b**) element content characteristics of rock sample.

**Figure 5 ijerph-19-13351-f005:**
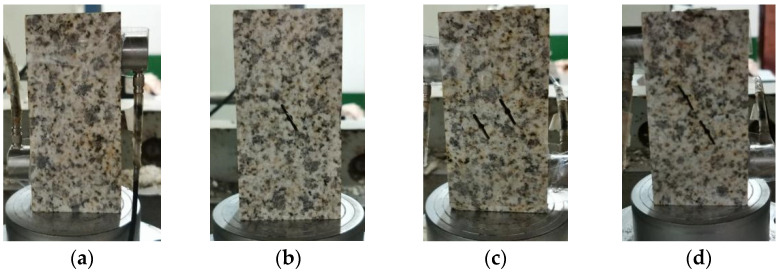
The rock testing machine. (**a**) The intact rock sample; (**b**) rock sample with a single crack; (**c**) rock sample with parallel double cracks; (**d**) rock sample with collinear double cracks; (**e**) rock sample with intermittent double cracks; (**f**) rock sample with conjugate double cracks; (**g**) rock sample with cross double cracks.

**Figure 6 ijerph-19-13351-f006:**
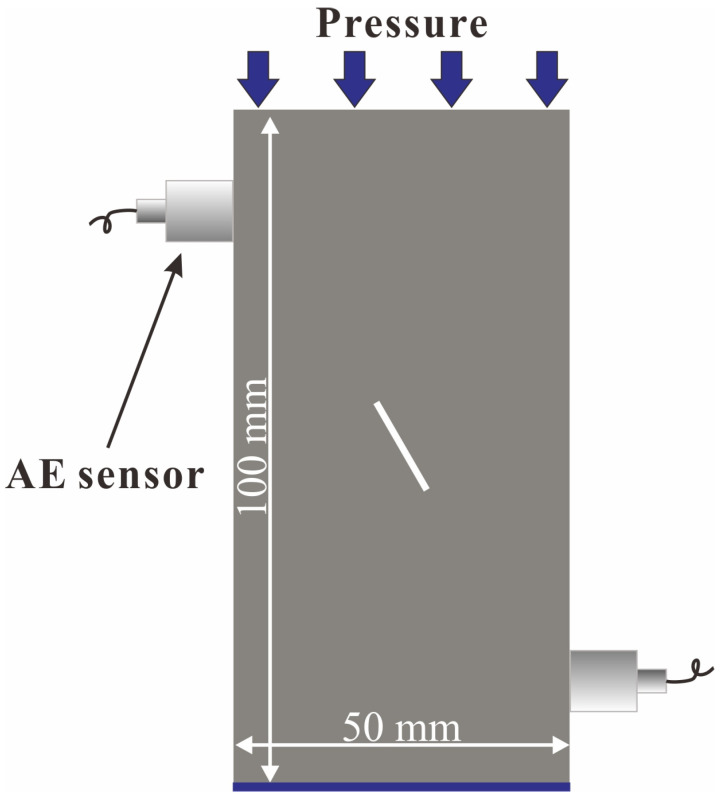
The boundary conditions of the model samples.

**Figure 7 ijerph-19-13351-f007:**
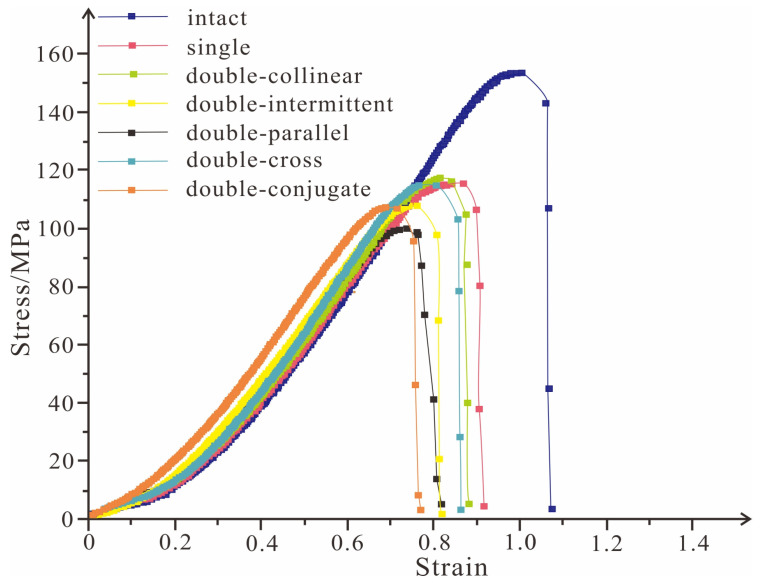
Strength of rock samples.

**Figure 8 ijerph-19-13351-f008:**
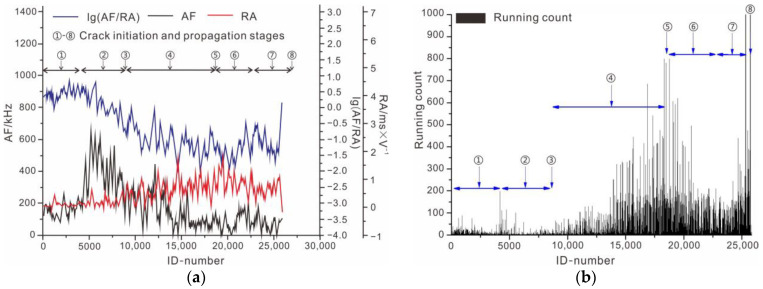
Parameters of acoustic emission of intact rock sample. (**a**) Curve of parameters of acoustic emission; (**b**) running count of intact rock sample.

**Figure 9 ijerph-19-13351-f009:**
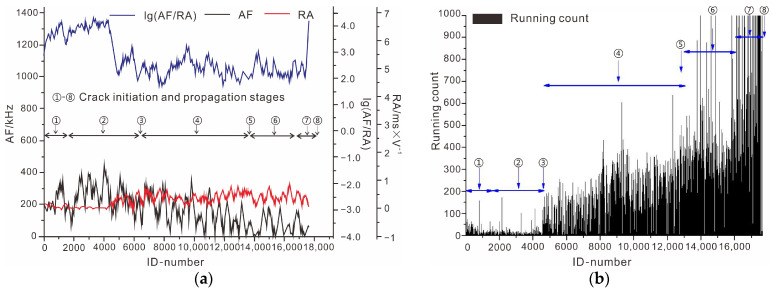
Parameters of acoustic emission of rock sample with single crack. (**a**) Curve of parameters of acoustic emission; (**b**) running count of intact rock sample.

**Figure 10 ijerph-19-13351-f010:**
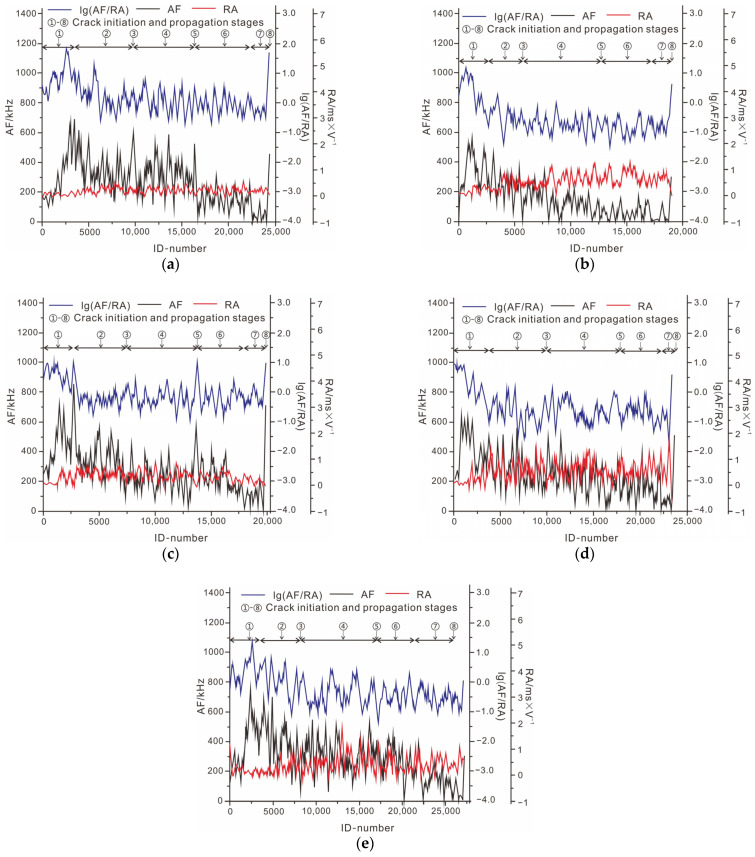
Parameters of the acoustic emission of rock sample with double cracks. (**a**) Curves of rock sample with conjugate double cracks; (**b**) curves of rock sample with collinear double cracks; (**c**) curves of rock sample with cross double cracks; (**d**) curves of rock sample with parallel double cracks; (**e**) curves of rock sample with intermittent double cracks.

**Figure 11 ijerph-19-13351-f011:**
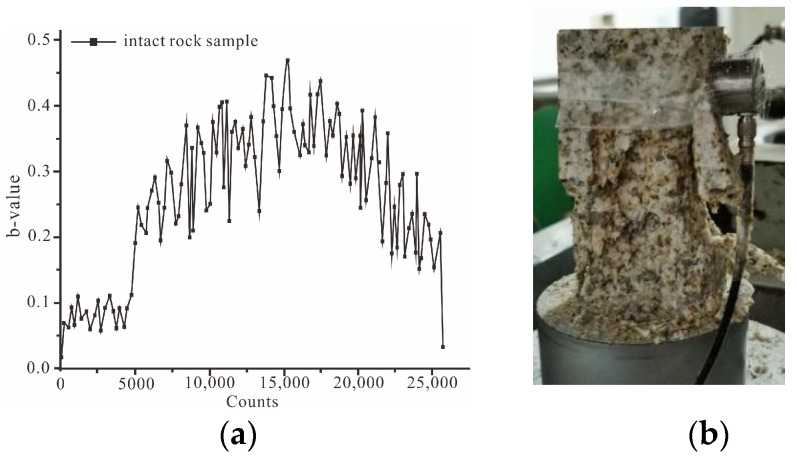
The *b*-value of intact rock sample. (**a**) The curve of the *b*-value of intact rock sample; (**b**) the failure of intact rock sample.

**Figure 12 ijerph-19-13351-f012:**
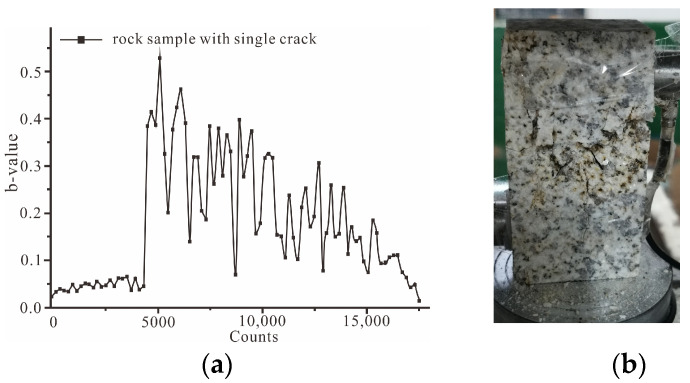
The *b*-value of the rock sample with a single crack. (**a**) The curve of the *b*-value of rock sample with single crack; (**b**) the failure of a rock sample with a single crack.

**Figure 13 ijerph-19-13351-f013:**
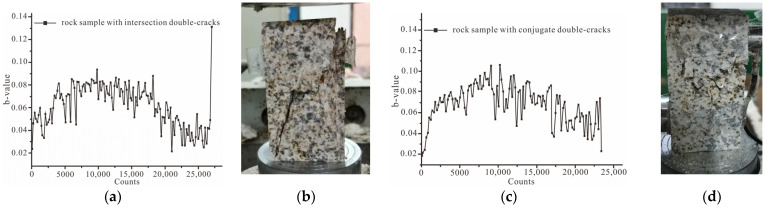
Parameters of the acoustic emission of rock samples with single cracks. (**a**) The curve of the *b*-value of rock samples with cross double cracks; (**b**) the failure of rock sample with cross double cracks; (**c**) the curve of the *b*-value of rock samples with conjugate double cracks; (**d**) the failure of rock samples with conjugate double cracks; (**e**) the curve of the *b*-value for rock samples with collinear double cracks; (**f**) the failure of rock samples with collinear double cracks; (**g**) the curve of the *b*-value of rock samples with parallel double cracks; (**h**) the failure of rock sample with parallel double cracks; (**i**) the curve of the *b*-value of rock sample with intermittent double cracks; (**j**) the failure of rock samples with intermittent double cracks.

## Data Availability

Not applicable.

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
