# Peer review of "Analysis of Acoustic Emission Characteristics and Failure Mode of Deep Surrounding Rock of Sanshandao Gold Mine"

_ijerph, 2022, doi:10.3390/ijerph192013351_

Round 1

Reviewer 1 Report

Title: Analysis of acoustic emission characteristics and failure mode of deep surrounding rock of Sanshandao Gold Mine   Authors: Li eat al.   In this manuscript, author have analyzed  acoustic emission and failure mode  of deep surrounding rock of Sanshandao Gold Mine in order to provide a data basis for deep surrounding rock stability in the study area, and gave suggestions for rock failure prediction in general.   The paper is original and well- organized and fit the scope of the IJERPH journal. However, I would like the authors to address major review comments as follow:   1- kindly, go through whole manuscript and correct English language and modify accordingly. And reduce repeated sentences.    2- What was the procedure for selecting the 7 samples ?    3-  the 7 samples have different cracks type except for the first sample “intact”, were these cracks naturally occurred in samples, or were they made after selecting samples ?    4- The details of crack in each sample is not clear. What is the size or  surface area or volume of crack in each sample. What is the % of crack area to the total area of the sample ?  These details are very important for this study.    5- the introduction section is short and provide no enough information especially for the acoustic emission background. I suggest author modify the introduction part and add more details and references.    6- the acoustic emission (AE) results for this study should be compared with other reported results. A table for comparing findings in this manuscript with other reported results should be crated.   7- in the discussion section, author make so many conclusion and assumptions based on the b-value curve for several samples. First, the b-value results for all samples should be summarized in a table. Secondly supportive references must be used after each conclusion made by author.   

Reviewer 2 Report

Now this manuscript is far away from acceptance, and the following comments should be considered in the revision process:

(1)    The introduction part is quite insufficient. Please enrich it by reviewing more milestone papers.

(2)    How many specimens do you prepare for each scenario? Please show the repeatability of the test results.

(3)    In Fig. 6, the stress did not start from 0. Why?

(4)    How do the authors identify crack types by AE signals?

(5)    This paper is much like a test report. Please explain the underlying mechanism of the experimental phenomenon.

(6)    Compared to the published papers in this subject, what is the most attracting difference and innovation of this study? Please show the reviewer and emphasize it in the manuscript.

(7)    Can this study enlighten the engineering? How can this be applied in field application?

(8)    The English is quite poor. I strongly suggest that the language should be greatly edited and polished by native speakers in this field.

Reviewer 3 Report

The study does not seem novel but just applied to a new area. As every rock has different properties, I would appreciate having mentioned the rock types when describing previous similar studies in the introduction. I would also recommend increasing the number of samples to produce more robust results. In the conclusion section, I miss information on how the study could be helpful in various fields, such as the mining industry.  

Round 2

Reviewer 1 Report

All review comments were answered properly. The revised manuscript has been improved greatly. I support publishing this manuscript in present form. 

Author Response

Thank you so much for your approval and help.

Reviewer 3 Report

Although the authors claimed so, they have not responded to many comments. For example, in Section 2.1, beresitization is not explained, mineralization is not described, and quaternary sediments are not described. The geological cross section is still missing. Therefore, I ask the author to provide answers to each of my comments one by one before I proceed with the review. Besides, for the figure, you should add an intermediate-scale regional map between the detailed geological map and the map of China (by the way, remove Taiwan from the map or label it properly). 
